# Natural Occurrence and Co-Occurrence of Beauvericin and Enniatins in Wheat Kernels from China

**DOI:** 10.3390/toxins16070290

**Published:** 2024-06-26

**Authors:** Wenjing Xu, Jiang Liang, Jing Zhang, Yan Song, Xi Zhao, Xiao Liu, Hongyuan Zhang, Haixia Sui, Jin Ye, Yu Wu, Jian Ji, Yongli Ye, Xiulan Sun, Jin Xu, Li Bai, Xiaomin Han, Lei Zhang

**Affiliations:** 1NHC Key Laboratory of Food Safety Risk Assessment, Chinese Academy of Medical Science Research Unit (2019RU014), China National Center for Food Safety Risk Assessment, Beijing 100021, China; xuwenjing@cfsa.net.cn (W.X.); liangjiang@cfsa.net.cn (J.L.); zhangjing@cfsa.net.cn (J.Z.); songyan@cfsa.net.cn (Y.S.); zhaoxi@cfsa.net.cn (X.Z.); liuxiao@cfsa.net.cn (X.L.); zhanghongyuan@cfsa.net.cn (H.Z.); suihaixia@cfsa.net.cn (H.S.); xujin@cfsa.net.cn (J.X.); baili@cfsa.net.cn (L.B.); 2National Food and Strategic Reserves Administration, Beijing 100834, China; yj@ags.ac.cn (J.Y.); wyu@ags.ac.cn (Y.W.); 3School of Food Science, State Key Laboratory of Food Science and Technology, National Engineering Research Center for Functional Foods, Jiangnan University, Wuxi 214122, China; jijian@jiangnan.edu.cn (J.J.); yyly0222@jiangnan.edu.cn (Y.Y.); sxlzzz@jiangnan.edu.cn (X.S.)

**Keywords:** emerging mycotoxins, contamination and co-contamination, wheat kernels, UPLC-MS/MS

## Abstract

A total of 769 wheat kernels collected from six provinces in China were analyzed for beauvericin (BEA) and four enniatins (ENNs), namely, ENA, ENA_1_, ENB and ENB_1_, using a solid phase extraction (SPE) technique with ultra-high performance liquid chromatography–tandem mass spectrometry (UPLC-MS/MS). The results show that the predominant toxin was BEA, which had a maximum of 387.67 μg/kg and an average of 37.69 μg/kg. With regard to ENNs, the prevalence and average concentrations of ENB and ENB_1_ were higher than those of ENA and ENA_1_. The geographical distribution of BEA and ENNs varied. Hubei and Shandong exhibited the highest and lowest positive rates of BEA and ENNs (13.46% and 87.5%, respectively). However, no significant difference was observed among these six provinces. There was a co-occurrence of BEA and ENNs, and 42.26% of samples were simultaneously detected with two or more toxins. Moreover, a significant linear correlation in concentrations was observed between the four ENN analogs (*r* range: 0.75~0.96, *p* < 0.05). This survey reveals that the contamination and co-contamination of BEA and ENNs in Chinese wheat kernels were very common.

## 1. Introduction

Beauvericin (BEA) and enniatins (ENNs) are structurally related cyclic hexadepsipeptides, which consist of alternating D-2-hydroxyisovaleric acid and N-methyl L-amino acids. Most of them are from the *Fusarium* species [1,2]. To date, more than 29 enniatin analogs have been reported. Only four of them are frequently detected in food and feed, namely, enniatin A (ENA), enniatin A_1_ (ENA_1_), enniatin B (ENB) and enniatin B_1_ (ENB_1_) [3]. Due to their ionophoric properties, BEA and ENNs can act as ion carriers through the plasma membrane or form a cation-selective channel, consequently altering cellular ionic homeostasis, thus leading to apoptosis and cytotoxicity [4,5]. In addition, BEA and ENNs show genotoxicity by producing DNA fragmentation, chromosomal aberrations and micronuclei [6]. In vivo studies have shown that the Lethal Dose 50 (LD50) for BEA and ENNs was 100 mg/kg b.w. and 350 mg/kg b.w. in mice after oral administration [7,8]. In vitro studies also reveal that BEA and ENNs show toxic effects on the reproductive system [8].

Unlike the well-known traditional Fusarium mycotoxins, there is a lack of data available on the toxicity and occurrence of BEA and ENNs. The limited data demonstrates that there is a common occurrence and co-occurrence of BEA and ENNs. The concentrations of BEA and ENNs range from a few μg/kg to several thousand mg/kg, and they have been found in wheat, barley, corn, oat, rice and their based products from different countries [9,10,11]. Therefore, these emerging mycotoxins have raised global concern due to their potential toxicity and high prevalence. However, no regulations or guidelines for BEA and ENNs have been set [12].

Given that wheat is the most consumed cereal worldwide, the natural occurrence of BEA and ENNs in wheat needs to be elucidated, especially in China, which is a major wheat producer and consumer. Thus, the object of this paper is to investigate the natural occurrence of BEA and ENNs in Chinese wheat kernels in order to give a better understanding of the co-contamination of these emerging mycotoxins in raw materials. The results obtained from our study will provide a scientific basis for assessing the impact of BEA and ENNs on the Chinese population resulting from the consumption of wheat products.

## 2. Results

### 2.1. Method Validation

The standard curves of the matrix-matched calibration obtained for BEA and ENNs showed excellent linearity (coefficient *r*^2^ > 0.99 for all curves) in the range of toxin-added levels. The LODs and LOQs for these five toxins in wheat kernels were 0.08 and 0.26 μg/kg for BEA, 0.06 and 0.21 μg/kg for ENA, 0.21 and 0.38 μg/kg for ENA_1_, 0.02 and 0.05 μg/kg for ENB, and 0.14 and 0.48 μg/kg for ENB_1_, respectively. The recoveries from triplicate samples at three spiked levels varied from 100.5% to 126.0% for BEA, 115.5% to 124.3% for ENA, 117.5% to 127.2% for ENA_1_, 114.6% to 123.1% for ENB, and 118.3% to 125.1% for ENB_1_, respectively. The repeatability and reproducibility of BEA and ENNs ranged from 0.1% to 6.8% and 3.9% to 13.3%.

### 2.2. Natural Occurrence of BEA and ENNs in Wheat Kernels

The contamination of BEA and ENNs in a total of 769 wheat kernels collected from China in 2018 and 2019 are summarized in Table 1. BEA was the predominant toxin in terms of concentrations in Chinese wheat. Forty-one samples (5.33%) presented detectable levels of BEA, with a mean value of 37.69 μg/kg (median = 7.80 μg/kg and maximum = 387.67 μg/kg). With respect to ENNs, nearly half of the samples (49.54%) were positive for at least one ENN. The total ENN concentrations (ENA + ENA_1_ + ENB + ENB_1_) varied from 0.04 μg/kg to 448.28 μg/kg, with an average and a median of 30.03 μg/kg and 9.19 μg/kg, respectively. Among the four ENNs, ENB showed the highest frequency at 46.68%, followed by ENB_1_ (40.83%), ENA (40.44%) and ENA_1_ (19.64%). However, regarding the concentrations of the four ENN analogs, they showed a different order. ENB_1_ was the most abundant toxin with a mean and a median level of 15.70 μg/kg and 5.48 μg/kg, respectively, and the maximum of ENB_1_ was 212.81 μg/kg. The average, median and maximum concentrations of the four ENNs from high to low were in the following order: ENB_1_ > ENB > ENA_1_ > ENA.

### 2.3. Geographical Distribution of BEA and ENNs in China

All these wheat kernels were collected from the six major wheat-producing provinces in China. The concentrations of BEA and ENNs in wheat kernels from different provinces are shown in Table 2. It was demonstrated that the natural occurrence of BEA and ENNs in Chinese wheat kernels varied geographically.

In terms of BEA contamination, the frequency of BEA in the analyzed samples from the highest to the lowest was in the following order: Hubei (13.46%) > Anhui (9.74%) > Henan (3.40%) > Jiangsu (2.22%) > Hebei (2.00%) > Shandong (1.27%). In addition, the mean and maximum levels of BEA in samples from Hebei (mean = 11.38 μg/kg and maximum = 19.58 μg/kg), Jiangsu (mean = 13.10 μg/kg and maximum = 33.50 μg/kg), Anhui (mean = 23.56 μg/kg and maximum = 165.84 μg/kg) and Hubei (mean = 25.54 μg/kg and maximum = 218.87 μg/kg) in the analyzed positive samples were all lower than the levels in the total positive wheat kernels. There was only one wheat sample contaminated with BEA (concentration = 179.67 μg/kg) in Shandong. The concentrations of BEA in Henan positive samples (mean = 55.42 μg/kg and maximum = 387.67 μg/kg) were higher than the concentrations in all the analyzed positive samples.

For ENNs, the incidence was the highest in the wheat samples from Hubei (87.50%), followed by Hebei (56.67%), Anhui (50.65%), Jiangsu (48.89%), Henan (37.41%) and Shandong (7.59%), and the average concentrations of ENNs in wheat kernels from Henan (42.91 μg/kg), Jiangsu (33.65 μg/kg) and Hubei (35.78 μg/kg) were higher than those in the total positive wheat samples. However, the mean levels of ENNs in the samples from Hebei (16.81 μg/kg), Shandong (0.6 μg/kg) and Anhui (27.87 μg/kg) were lower than those in all the analyzed wheat kernels.

Figure 1 also shows that the ENN concentrations were lower in the samples from Hebei and Shandong. However, further statistical analysis indicates that there was no significant difference in BEA or ENN concentrations among wheat samples from these six provinces.

### 2.4. Co-Occurrence of BEA and ENNs in Chinese Wheat Kernels

The co-occurrence of BEA and ENNs in wheat kernels collected from China is shown in Figure 2. The co-contamination of BEA and ENNs in Chinese wheat kernels was quite frequent. At least one toxin was detected in over 50% of wheat samples. Specifically, there were 8.19% of wheat samples contaminated by one toxin, 5.20% by two toxins, 16.12% by three toxins, 18.60% by four toxins, and 2.34% by five toxins. The co-contamination mainly occurred among ENNs. The composition of ENNs is shown in Figure 3. It was found that the concentrations of individual toxins in total ENNs exhibited the same characteristics in Chinese wheat kernels from different provinces, and ENB and ENB_1_ were more abundant than ENA and ENA_1_. Specifically, ENB_1_ and ENB accounted for 42.36% and 40.56% of the total ENNs, respectively, which is over 80% when added together, while ENA_1_ accounted for a fraction below 15%, and ENA was less than 5%. Moreover, a significant linear correlation in the concentrations was observed between ENA and ENA_1_ (*r* = 0.754, *p* < 0.05), ENA and ENB (*r* = 0.858, *p* < 0.05), ENA and ENB_1_ (*r* = 0.896, *p* < 0.05), ENA_1_ and ENB (*r* = 0.75, *p* < 0.05), ENA_1_ and ENB_1_ (*r* = 0.794, *p* < 0.05) and ENB and ENB_1_ (*r* = 0.96, *p* < 0.05) (Table 3). This may imply that the production of these four ENNs was homologous.

## 3. Discussion

The Food and Agriculture Organization of the United Nations (FAO) issued a statement in 2022 that considered cereals a staple food, accounting for about one-third of total diets [13]. Wheat, as a leading cereal, is widely planted and largely consumed in daily diets. Although wheat contributes to the nourishment of humankind, inevitable contamination by fungi and mycotoxins has become an emerging concern for public health. The fungi of the Fusarium species are the most important seed-borne contaminants in wheat associated with diversified mycotoxins, such as deoxynivalenol and zearalenone. Among these mycotoxins, a group of bioactive compounds, BEA and ENNs, is neither routinely determined nor legislatively regulated, and these compounds are also called emerging or minor mycotoxins [14]. In recent years, the incidence of BEA and ENNs has been described as frequent in food and feed from several countries, and the concentrations of BEA and ENNs were detected with unacceptable levels of up to 10~500 mg/kg [14]. These facts have attracted more and more attention from the scientific community. Since there is little data about the presence of BEA and ENNs in the unprocessed grains of China, our study aims to determine BEA and ENNs present in wheat kernels for human consumption in China.

The characteristics of BEA and ENN contamination found in the present study are in accordance with those described by the European Food Safety Authority (EFSA), in which BEA was found in 9~79% of wheat samples, and the maximum concentrations of BEA were usually in the μg/kg range. The prevalence and concentration were also found to be the highest for ENB and ENB_1_ and the lowest for ENA [3]. When compared with the results reported in other countries, for example, the frequency of BEA (5.33%) and ENNs (19.64~46.68%) detected in Chinese wheat kernels was higher than those in Romanian wheat in the extremely wet year of 2014 and extremely dry year of 2015 (2% and 8~41%), and the average levels of BEA obtained in our study were higher than those reported in Romania (37.69 μg/kg vs. 0.07 μg/kg). In contrast, the mean concentrations of ENNs in Chinese wheat kernels were lower than those detected in Romanian wheat (1.00 μg/kg vs. 4.5 μg/kg for ENA, 5.33 μg/kg vs. 13.7 μg/kg for ENA_1_, 15.03 μg/kg vs. 54.8 μg/kg for ENB and 15.70 μg/kg vs. 30.5 μg/kg for ENB_1_) [15]. Similar results were found in comparison with those reported in Norwegian wheat in the growing seasons from the years 2000 to 2002, and the average levels of BEA and ENNs were lower than 3.0 μg/kg, 5.8 μg/kg, 22 μg/kg, 790 μg/kg and 180 μg/kg, respectively [16]. According to the Koppen–Geiger climate classification system, the wheat-growing regions in Romania have a humid temperate continental climate (Dfa and Dfb), and in Norway, they have an oceanic climate (Cfb), while, in China, we have a humid subtropical climate (Cfa). Different climate conditions will affect wheat growth and toxin production [17]. Moreover, it was reported by Han et al. [18] that the average levels of BEA and the four ENNs in wheat samples from Shandong province in China were 0.68 μg/kg, 0.87 μg/kg, 5.33 μg/kg, 12.55 μg/kg and 15.61 μg/kg, which were higher those found in this study.

The geographical differences in BEA and ENN contaminations in Chinese wheat kernels are also described in this paper. As we know, climate conditions (especially temperature and precipitation) during the wheat flowering and grouting stage are critical for wheat growth. These six wheat sampling provinces from high to low in terms of latitude are successively Hebei, Shandong, Henan, Jiangsu, Anhui and Hubei. According to the Köppen–Geiger climate classification system, Hebei belongs to the arid zone and cold steppe; Shandong is located in the cold zone, and it is warm in summer and dry in winter; Henan and Jiangsu are both distributed in the temperate zone, and their summers are dry and hot; and Anhui and Hubei also belong to the temperate zone, where the summers are hot, but there is no dry season [19]. In general, the hot and humid weather during the wheat flowering and grouting seasons facilitates the toxin production of the *Fusarium* species [20]. Although the climate conditions vary so much in these areas, there was no significant difference in BEA or ENN contamination. This might be as a result of agricultural management, which also plays an important role in the quality and yield of wheat. Using comprehensive and targeted measures, there seems to be “no difference”.

This paper also demonstrates that there was a common co-occurrence of BEA and ENNs in the analyzed samples. Stanciu et al. also summarized that 28% of the analyzed wheat samples were contaminated with at least two BEA and ENNs [15]. Additionally, a significant linear correlation was observed between the four ENNs contaminating Chinese wheat kernels. Since ENNs are of a similar chemical structure, the biosynthesis of these compounds might share a common pathway. To our knowledge, ENNs can be produced efficiently by strains of numerous *Fusarium* species in vitro and in planta, such as *Fusarium oxysporum*, *F. poae* and *F. avenaceum*. Some *Fusarium* species have been reported to produce different ENN analogs simultaneously [21]. However, certain ENNs can be isolated from only a particular species or strain of *Fusarium*. For example, *F. sambucinum* preferably produces ENA, while *F. scirpi* preferably synthesizes ENB [22]. In the *Fusarium* species, ENN production is catalyzed by a non-ribosomal multifunctional enzyme called enniatin synthetase (ESYN1), which is encoded by the *esyn1* gene [23]. The biosynthesis of ENNs is regulated not only by genes but also by environmental factors in the field, including climate, agricultural practices and host plants [24]. For example, Brennan et al. found that the optimum temperature for the growth of *F. graminearum*, *F. culmorum* and *F. poae* was 25 °C, while, for *F. avenaceum*, it was 20 °C [25]. The interactions between these factors determine the type and amount of ENNs.

This survey reveals the natural occurrence of BEA and ENNs in unprocessed wheat samples from China. A better understanding of the fate of BEA and ENNs during the subsequent processing of wheat is that milling, sourdough fermentation, malting and baking can lead to a decrease in these mycotoxins. It was reported by Hu et al. that after the milling of wheat grains naturally contaminated with ENB and ENB_1_, approximately 70% to 82% of the two ENNs were found in the bran fraction, and the rest remained in the flour. During sourdough fermentation, BEA and ENNs contents can be reduced, in particular at 40 °C, and baking at 200 °C for 25 min leads to a further decrease in BEA and ENNs, ranging from 9% to 28% compared with fermented dough [26]. Vaclavikova et al. also observed a large drop in ENN concentrations in the brewing process of beer [27]. This may also suggest that the occurrence of BEA and ENNs in raw materials does not result in a risk of exposure to products based on them. Additionally, high levels of BEA and ENNs in unprocessed grains might be accumulated in feed, which should be cause for concern.

Cereal and cereal-based products, especially wheat and wheat-based products, are the largest contributors to BEA and ENN exposure. In addition, the Chinese population registers high wheat and wheat product consumption. Therefore, there is a need to estimate the exposure to BEA and ENNs from wheat and wheat-based products in Chinese populations. The present study comprises an extensive survey about BEA and ENNs in Chinese wheat, and more targeted studies on the toxicity and occurrence of BEA and ENNs are needed for risk assessments so that developing pertinent strategies to prevent and mitigate these emerging mycotoxins can be achieved.

## 4. Conclusions

This study demonstrates that the contamination and co-contamination of BEA and ENNs in Chinese wheat kernels were very common. Among five emerging mycotoxins, it was found that BEA was the predominant toxin, followed by ENB, ENB_1_, ENA and ENA_1_, and over two-fifths of samples were simultaneously detected with at least two toxins. Moreover, there was a significant linear correlation in concentrations observed between the four ENN analogs. All these findings suggest that more data and routine surveillance are needed to further assess the risk of these emerging mycotoxins.

## 5. Materials and Methods

### 5.1. Chemicals and Reagents

Acetonitrile and methanol (both LC-MS grade) were purchased from Fisher Scientific (Fair Lawn, NJ, USA). Ammonium acetate was of MS grade from Sigma-Aldrich (St. Louis, MO, USA). Pure water was obtained from a Millipore Milli-Q Water Purification System (Millipore, Bedford, MA, USA) with conductivity ≥18.2 MΩ·cm and total organic carbon (TOC) < 5 µg/L at 25 °C. Standard powders of BEA, ENA, ENA_1_, ENB and ENB_1_ with purities > 97% were purchased from BioAustralis (Smithfield, NSW, Australia).

### 5.2. Sample Collection

A total of 769 wheat kernels were collected from six major wheat-producing provinces of China, namely, Anhui, Hubei, Hebei, Henan, Jiangsu and Shandong during the 2018 and 2019 harvest seasons. The sampling provinces are shown in Figure 4. After being harvested, dried and threshed, fresh wheat kernels were sampled directly from the farmers’ fields, which were randomly selected and designated by the Academy of National Food and Strategic Reserves Administration in China. Each wheat kernel sample weighed at least 2.5 kg. Then, a random quarter of the sample was selected, and half was ground to a fine powder of 20 meshes using a Blender 8010ES (Warning Commercial, Torrington, CT, USA). All finely ground samples were kept in Ziploc plastic bags at 4 °C prior to analysis.

### 5.3. Toxin Analysis

An ultra-high performance liquid chromatography coupled with mass spectrometry (UPLC-MS/MS) method was employed to detect BEA, ENA, ENA_1_, ENB and ENB_1_ in wheat kernels. Briefly, a finely ground powder (5 g) was mixed with 40 mL acetonitrile–water (85:15, *v*/*v*) and blended for 60 s by a vortex, followed by horizontally shaking for 30 min at 180 rpm. The mixture was left to stand for 15~20 min at room temperature. An aliquot of 10 mL supernatant was transferred to a 50 mL centrifuge tube, diluted with 20 mL pure water, mixed by a vortex and allowed to stand for 15~20 min at an ambient temperature. A portion of 4 mL diluent was loaded on a solid phase extraction (SPE) column, which was pre-equilibrated by 3 mL methanol and 3 mL water. The column was washed with 3 mL of acetonitrile–water (10:90, *v*/*v*), followed by 3 mL of acetonitrile–water (50:50, *v*/*v*). Toxins were eluted with 2 mL of acetonitrile–water (90:10, *v*/*v*), analyzed and confirmed by UPLC-MS/MS based on a previous method in our lab [28]. The matrix-matched calibration standard curve was employed for quantification.

### 5.4. UPLC MS/MS Conditions

The qualitative and quantitative analysis of BEA and ENNs was simultaneously achieved by an UPLC-MS/MS system, which was equipped with Exion LC (Shimadzu, Kyoto, Japan) and QTrap 5500 MS/MS (AB Sciex, Foster City, CA, USA). For chromatographic separation, a C_18_ column (2.1 mm × 50 mm, 1.7 μm bead diameter, Waters, Milford, MA, USA) was used. A binary gradient with mobile phase A (2 mmol/L ammonium bicarbonate) and B (acetonitrile) was programmed, starting at A of 100%, which was maintained for the first 2 min. Then, A declined to 40% for 1 min. Afterward, A linearly decreased to 30% for 16 min and increased to 100% for the following 2 min. The flow rate was set to 0.2 mL/min.

The MS/MS was performed in the mode of positive electrospray ionization (ESI^+^) with multiple reaction monitoring (MRM) for BEA and ENN detection. The MS source-dependent parameters were optimized and displayed as follows: ion spray voltage to 5500 V, source temperature to 550 °C, gas1 to 80 psi, gas2 to 80 psi, curtain gas to 30 psi and collision-activated dissociation gas to a medium level.

### 5.5. Method Validation

The method proficiency was validated for linearity, limit of detection (LOD), limit of quantification (LOQ), recovery, repeatability and reproducibility. For matrix-matched calibration, standard solutions of BEA and ENNs were added to toxin-free wheat extracts at levels between 0.03 and 100 μg/L for BEA and ENB_1_, 0.005 and 15 μg/L for ENA, and 0.01 and 400 μg/L for ENA_1_ and ENB, respectively. The LOD and LOQ were determined based on the signal-to-noise ratios of 3:1 and 10:1, respectively. To evaluate the recovery, toxin-free wheat samples were spiked with BEA and ENB_1_ at the concentrations of 50, 100 and 250 μg/kg, with ENA at the concentrations of 7.5, 15 and 37.5 μg/kg, and ENA_1_ and ENB at the concentrations of 20, 40 and 100 μg/kg, respectively. Triplicates of each concentration were tested. Then, a spiked wheat kernel sample with the analyzed five toxins at the concentrations of 10 μg/kg for BEA and ENB_1_, 1.5 μg/kg for ENA and 4 μg/kg for ENA_1_ and ENB was analyzed six times in one day and once a day on five successive days for repeatability and reproducibility calculations, respectively.

### 5.6. Data Analysis

Significant differences in BEA and ENNs concentrations were tested using the Kruskal–Wallis H test or Mann–Whitney U test. The relationships of concentrations between any of the five emerging mycotoxins were tested using the Spearman correlation. All statistical analyses were computed using the SPSS statistical package (version 20.0, IBM, Armonk, NY, USA).

## Figures and Tables

**Figure 1 toxins-16-00290-f001:**
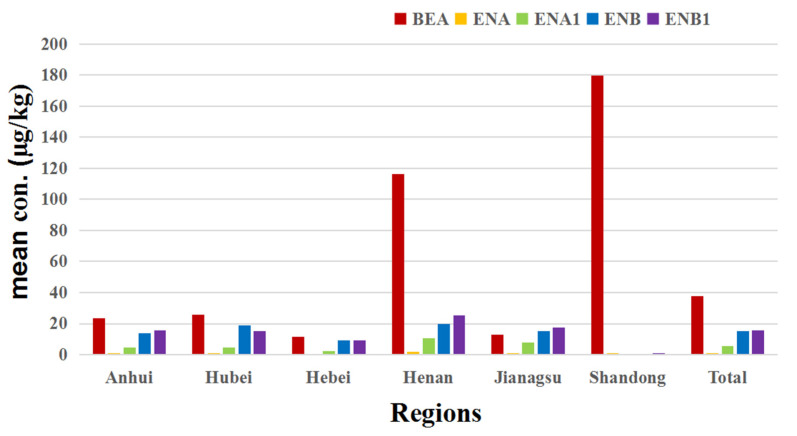
Mean concentrations of BEA and ENNs in wheat kernels from six provinces in China in the years 2018 and 2019.

**Figure 2 toxins-16-00290-f002:**
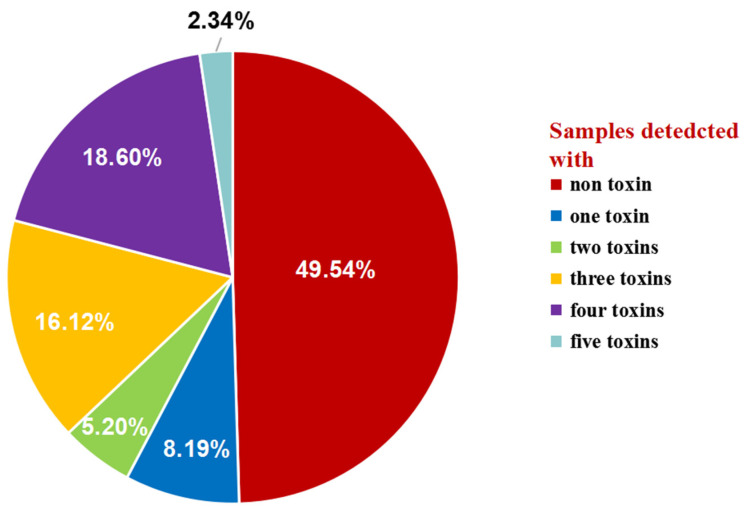
Occurrence and co-occurrence of BEA and ENNs in Chinese wheat kernels.

**Figure 3 toxins-16-00290-f003:**
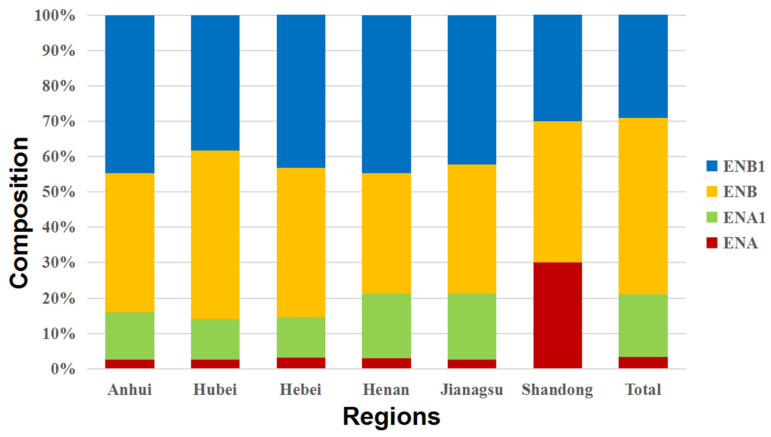
The composition of the four ENNs in wheat kernels from six provinces in China in the years 2018 and 2019.

**Figure 4 toxins-16-00290-f004:**
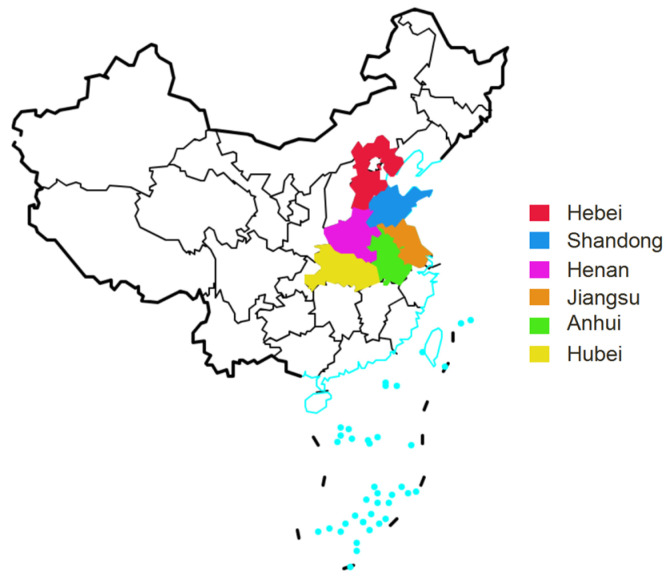
A map of China where wheat kernels were sampled from provinces in the years 2018 and 2019.

**Table 1 toxins-16-00290-t001:** The natural occurrence of BEA and ENNs in wheat kernels collected from China in 2018 and 2019.

Mycotoxin	Range (μg/kg)	Average (μg/kg)	Median (μg/kg)	Positive (%)
BEA	0.09~387.67	37.69	7.80	5.33 (41/769)
ENA	0.06~10.77	1.00	0.37	40.44 (311/769)
ENA_1_	0.21~38.00	5.33	2.39	19.64 (151/769)
ENB	0.04~187.79	15.03	4.90	46.68 (359/769)
ENB_1_	0.14~212.81	15.70	5.48	40.83 (314/769)
ENNs	0.04~448.28	30.03	9.19	49.54 (381/769)

**Table 2 toxins-16-00290-t002:** Natural occurrence of BEA and ENNs in wheat kernels from different provinces in China in 2018 and 2019.

Province	Mycotoxin	Range (μg/kg)	Average (μg/kg)	Median (μg/kg)	Positive (%)
Hebei	BEA	3.57–19.58	11.38	10.99	2.00 (3/150)
	ENA	0.07–10.77	0.67	0.37	52.00 (78/150)
	ENA_1_	0.24–12.02	2.49	1.31	21.33 (32/150)
	ENB	0.04–120.82	9.15	4.02	48.67 (73/150)
	ENB_1_	0.18–93.08	9.39	1.87	44.67 (67/150)
	ENNs	0.04–228.02	16.81	4.69	56.67 (85/150)
Shandong	BEA	-	179.67	179.67	1.27 (1/79)
	ENA	0.09–1.19	0.40	0.16	5.06 (4/79)
	ENA_1_	-	-	-	0.00 (0/79)
	ENB	0.14–1.09	0.53	0.36	3.80 (3/79)
	ENB_1_	-	0.40	0.40	1.27 (1/79)
	ENNs	0.09–1.49	0.6	0.3	7.59 (6/79)
Henan	BEA	0.09–387.67	116.21	55.42	3.40 (5/147)
	ENA	0.06–9.90	1.74	0.32	27.21 (40/147)
	ENA_1_	0.21–28.78	10.43	8.64	12.93 (19/147)
	ENB	0.08–137.84	19.62	3.45	35.37 (52/147)
	ENB_1_	0.14–146.71	25.51	5.37	28.57 (42/147)
	ENNs	0.08–320.22	42.91	3.76	37.41 (55/147)
Jiangsu	BEA	1.98–33.50	13.10	3.83	2.22 (3/135)
	ENA	0.06–9.69	1.06	0.31	40.74 (55/135)
	ENA_1_	0.23–38.00	7.67	3.77	17.04 (23/135)
	ENB	0.07–187.79	15.08	4.35	48.15 (65/135)
	ENB_1_	0.37–212.81	17.35	4.43	42.96 (58/135)
	ENNs	0.06–448.28	33.65	7.85	48.89 (66/135)
Anhui	BEA	0.13–165.84	23.56	7.19	9.74 (15/154)
	ENA	0.06–9.38	0.89	0.38	37.66 (58/154)
	ENA_1_	0.27–35.94	4.61	2.39	21.43 (33/154)
	ENB	0.22–134.32	13.67	4.64	48.70 (75/154)
	ENB_1_	0.44–162.09	15.49	6.57	39.61 (61/154)
	ENNs	0.06–341.73	27.87	7.86	50.65 (78/154)
Hubei	BEA	0.59–218.87	25.54	8.34	13.46 (14/104)
	ENA	0.07–8.84	1.03	0.52	73.08 (76/104)
	ENA_1_	0.39–28.29	4.50	2.53	42.31 (44/104)
	ENB	0.37–125.84	18.68	10.18	87.50 (91/104)
	ENB_1_	0.20–129.37	15.05	7.03	81.73 (85/104)
	ENNs	0.50–287.48	35.78	13.11	87.50 (91/104)

**Table 3 toxins-16-00290-t003:** Spearman correlation coefficient between the four ENNs concentrations in wheat kernels in China in the years 2018 and 2019.

Spearman Correlation Coefficient	ENA	ENA_1_	ENB	ENB_1_
ENA	1	0.745	0.858	0.896
ENA_1_	0.745	1	0.75	0.794
ENB	0.858	0.75	1	0.96
ENB_1_	0.896	0.794	0.96	1

## Data Availability

The data that support the findings of this study are available from the corresponding author upon reasonable request.

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
