# Peer review of "Natural Occurrence and Co-Occurrence of Beauvericin and Enniatins in Wheat Kernels from China"

_toxins, 2024, doi:10.3390/toxins16070290_

Round 1

Reviewer 1 Report

Comments and Suggestions for Authors

The authors have carried out an extensive study on the toxicity of BEA and ENNS in several provinces in China.  Their study highlights the need to consider these emerging mycotoxins because of their occurrence in wheat and potential to cause apoptosis and cytotoxicity in man.   The manuscript will be of interest to environmentalists, health professionals and the farming industry.  The authors should consider the following comments.

2. Results para 1 Ln1 English "this section may be divided by sub-headings" change to "are divided" also on Ln 1 It "should" change to" It provides"

para2. Ln 6 delete "were"

The authors should check the manuscript for minor English errors..

Figure 1 Enlarge this figure as the labelling on the X-axis is too small

Figure 2 the labelling in this figure is ambiguous e. g.  three toxins or one toxins -what does this mean?  Either the labelling or the caption needs to be improved.

figure 3 This figure needs to be enlarged particularly the labelling on the x-axis.

Discussion The authors should add some comments on why the concentration of BEA and ENNS vary so much in the different regions they studied.

Reviewer 2 Report

Comments and Suggestions for Authors

  The manuscript toxins-3026681 presents the natural co-occurrence of the mycotoxins beauvericin (BEA) and enniatins (ENA, ENA1, ENB, ENB1) produced by Fusarium spp. in 769 samples of common wheat collected in six regions of eastern China. Wheat samples were analyzed by SPE and UHPLC-MS/MS methods, followed by SPSS statistical analysis. The results obtained are presented in detail, highlighting a geographical distribution of mycotoxins in wheat samples but without significant differences between provinces, as well as the coexistence and significant statistical correlations between BEA and ENNs.

I recommend the following improvements before publishing.

Please see the Review Report. 

Comments on the Quality of English Language

Minor editing of the English language required

Round 2

Reviewer 2 Report

Comments and Suggestions for Authors The authors have properly addressed the suggestions made, but I recommend the following improvements before publication.

L20. Key Contribution: This survey revealed a common occurrence and co-occurrence of BEA and ENNs in unprocessed wheat samples from eastern China.

L66. The contamination of BEA and ENNs in a total of 769 wheat kernels collected from  China in 2018 and 2019 are summarized in Table 1.

Table 1. The natural occurrence of BEA and ENNs in wheat kernels collected from China in 2018 and 2019

Table 2. Natural occurrence of BEA and ENNs in wheat kernels from different provinces of China in 2018 and 2019

L85 - 95, L96 - 102 and L103 - 105.  Separated paragraphs for better visualisation.

L161—163. When compared with the results reported in other countries, for example, the frequency of BEA (5.33%) and ENNs (19.64% ~ 46.68%) detected in Chinese wheat kernels was higher than those in Romanian wheat in the extremely wet 2014 and extremely dry 2015 years (2% and 8% ~ 41%).

Chapter 4.2. Please clarify the sampling. It is not clear to me from which samples the mycotoxin analysis was performed:

- 769 wheat kernels (approx. 1 kg ???) or

- 769 samples of wheat kernels (769 x approx. 1 kg ???)

- what weight/sample???

*** I strongly recommend using a map of China with the names of the provinces, to highlight the eastern wheat sampling regions. There are free online maps.  

1 Kernel is this:  https://www.ndsu.edu/faculty/simsek/wheat/kernel.html

Comments on the Quality of English Language

Minor editing of English language required.
